# Numerically Evaluation of Dynamic Behavior of Post-Tensioned Concrete Flat Slabs under Free Vibration

**Faham Tahmasebinia \*** **, Zhiyuan Hu, Qianhao Wei and Wenjie Ma**

School of Civil Engineering, University of Sydney, Darlington, Sydney 2006, Australia
* Correspondence: faham.tahmasebinia@sydney.edu.au

**Abstract:** The objective of this paper is to investigate the dynamic behaviour of post-tensioned concrete flat slabs with different geometries and damping ratios. Four groups of models with different lengths, widths, thicknesses and damping ratios designed according to the AS3600 standard. These were used to determine the influence of each parameter on the vibration serviceability by comparing the control variable method with the reference model. The vibration assessment parameters were used as natural frequency, peak acceleration, and response factor. Both the SCI/CSTR43 standard theoretical calculations method and the Strand7 finite element analysis (FEA) method are used to determine the effect of different geometries and damping ratios on vibration. The feasibility of the Strand7 FEA method for vibration analysis is also assessed by calculating the errors of the two methods. The paper concludes that the Strand7 FEA method is highly accurate and feasible. The span in both directions has a large effect on the natural frequency, and increasing both the slab thickness and the damping ratio are effective methods to improve the vibration serviceability. Based on the research in this paper, recommendations are provided for future vibration design of post-tensioned concrete slabs.

**Keywords:** post-tensioned concrete flat slabs; pedestrian loading; finite element analysis; theoretical calculations

## 1. Introduction

With urbanisation and the popularity of high-rise buildings, prestressed concrete floor systems have become widely used in high-rise buildings to satisfy the design demands of the modern large-span spatial structures. However, due to the large spans and low support resulting in the high flexibility and low natural frequency of prestressed concrete floor, it is susceptible to more significant vibrations under the action of human activity [1]. The vibrations will cause discomfort and psychological panic to the occupants, and severe vibrations can cause fatigue damage to the slab [2]. For the office and residential areas of high-rise buildings, pedestrian loads are the main form of vibration excitation in the building slab. As pedestrian loads are frequent and inevitable, vibration from pedestrian loads is an important aspect of vibration comfort research. Currently, most of the pedestrian load vibration studies are focused on footbridges and composite floor system, and the research on vibration serviceability of post-tensioned concrete floor is an important issue to be addressed nowadays.

Vibration occurs when a flooring system is subjected to pedestrian loads. It has been found that resonance occurs when the vibration frequency of the pedestrian load is close to the natural frequency of the floor system [3]. Therefore, the natural frequency and the forced vibration frequency are important factors to study when investigating the vibration serviceability of post-tensioned concrete floors. According to Howarth and Griffin [4] studies, the geometry of the structure, the material properties, the method of structural connection, and the damping all affect the vibration of the structure.

The main objective of this paper is to investigate the effect of post-tensioned concrete flat slab geometry and damping ratios on vibration comfort using finite element analysis. In

the paper, post-tensioned concrete flat slab models of different lengths, widths, thicknesses and damping ratios are designed using the AS3600-2018 specification and analysed by finite element modelling using Strand7. A comprehensive vibration serviceability assessment of post-tensioned concrete flat slabs is summarised based on the current international vibration comfort criteria, and obtain specific vibration serviceability assessment parameters and criteria. These vibration assessment parameters are calculated through SCI/CSTR43 standard calculations and the Strand7 finite element analysis, and comparisons are made to determine the feasibility and accuracy of the Strand7 FEA method. Finally, the paper will obtain the influence of length, width, thickness and damping ratio on the vibration comfort of post-tensioned concrete flat slab and give some design recommendations.

*1.1. Literature Review*

1.1.1. The Source of Vibration—Simulation of Pedestrian Loads

The human single-foot drop load is a fundamental component of the pedestrian load model. Harper [5] was the first to perform a test of the walking curve, proposing a single-foot drops M-shaped forcing function curve. It is the fundamental model that provides the methodology for the investigation of complex loads. Galbraith and Barton [6] systematically investigated the factors influencing walking vertical loads. It was shown that the peak vertical load was mainly influenced by pedestrian weight and stride frequency, with shoe type and ground conditions having less influence on the peak load. Van Nimmen, Lombaert [7] used three-dimensional inertial motion tracking to simulate walking load and acceleration responses in the field, verifying M-shaped forcing function curves for landing feet and time superposition curves for continuous walking loads. While pedestrian load curves can be obtained by experimental measurement methods, the measured curves do not facilitate engineering design applications therefore mathematical simplification is required.

Murray, Allen [8] tested the walking load profile using an inspection platform and showed that the load profile can be expressed as a sine function. Currently, the single and multi-person walking load model $F_w$ is commonly expressed as a Fourier function in Equation (1):

$$F_w(t) = G[1 + \sum_{i=1}^{n} a_{iw} \sin(2\pi i f_w t - \theta_{wi})] \tag{1}$$

where:
$G$ = weight of the pedestrian;
$n$ = total number of harmonics;
$a_{iw}$ = dynamic amplification factor of the i-th order;
$t$ = time of walking;
$f_w$ = walking frequency;
$\theta_{wi}$ = phase angle of the i-th order of the walking frequency.

Ellingwood and Tallin [9] found by experimental analysis that the first three harmonics of the pulse are the most important. The average relative amplitude of the first harmonic is the largest, the average relative amplitude of the next two harmonics is less than half of the amplitude of the first harmonic, and the fourth and higher harmonics are less than 10% of the first harmonic. Rainer, Pernica [10] compared the excitation of people at different step frequencies and concluded that only the fourth order harmonic number is needed to represent the pedestrian excitation load. The Fourier coefficients for walking curve. Kerr [11] sampled and analysed a large number of pedestrian excitation forces and this analysis demonstrated a very clear linear relationship between the 1st order harmonic factor and the step frequency, while the higher order harmonic factors showed a large discrete pattern.

1.1.2. The Transmission Pathway—Study of the Dynamic Properties of the Floor System

The natural frequency is the key parameter related to the floor to produce excessive vibration. Allen and Pernica [12] found that since the natural frequency of the majority of

floors is usually between 4 Hz and 8 Hz, greater than the first harmonic frequency. Thus, the vibration is probably caused by the second and third harmonics. They concluded from experiments that the lower the harmonic, the greater the vibration generated by the resonance. Pate and Built [13] found that for floor systems with the natural frequency of less than four times the pedestrian load frequency, the pedestrian load lead to an accumulation of vibrations, which is referred to as resonant excitation. For floors with natural frequency higher than four times the pedestrian load frequency, the response is mainly generated by the impulse load corresponding to the heel strike, with the vibration being a transient response to each individual impact of the pedestrian load.

Damping suppresses vibration through the energy dissipation. An increase in the level of damping allows more forced vibration energy to be dissipated in the form of sound and heat, resulting in lower levels of structural vibration [14]. Siringoringo, Abe [15] found that the damping of the floor is difficult to determine accurately due to the dense floor vibration modes caused by the orthogonal anisotropy of the bending capacity of the floor. Experience with similar building floors would therefore provide a reasonable estimate of the amount of damping.

### 1.1.3. The Vibration Receivers—Human Response to Vertical Vibration of the Floor System

Reiher [16] conducted the first research on the comfort of human vibration. In this experiment, participants were subjected to continuous vertical and horizontal steady state vibrations while in a seated and tilted position. They investigated the relationship between vibration frequency and amplitude in different environments and obtained that vibration comfort was determined by vibration speed based on the relationship curves. The experiments were the first to develop a general description of the vibration response and to discover an approximate law of comfort frequencies. Dieckmann [17] proposed an intensity perception parameter KB in conjunction with psychophysics and was used in the German standard DIN 4025. This standard firstly presents different criteria for assessing comfort in different environments and subdivides them by place and time of usage.

Hicks [3] suggested that human perception of vibration is also influenced by frequency and duration, and to avoid inaccurate vibration results from selecting unrepresentative peaks in continuous or transient responses, root mean square (RMS) acceleration values are commonly used and can be calculated using Equation (2).

$$a_{rms} = (\frac{1}{T} \int_0^T a(t)^2 dt)^{\frac{1}{2}} \tag{2}$$

where:

$a(t)$ = the time-history acceleration

$T$ = the time period

The BS6472 [18] standard introduces the concept of a vibration perception base curve. The base curve has the horizontal coordinate of frequency and the vertical coordinate of acceleration root-mean-squared and it shows the level of vibration that can be perceived by humans at different frequencies. The base curve was later applied to the ISO 2613 standard as a limiting curve for acceleration by multiplying the base line by the corresponding numerator for different environments.

### 1.1.4. The Main Criteria and Standards for Vibration Comfort of the Floor System

At present, there are two main methods to assess vibration comfort of large-span building floor: the frequency adjustment method, that the natural frequency of the building slab is higher than a certain limit value; and the limit dynamic response value method, that the limit value of the dynamic response of the building slab under the load. The three commonly used international standards and limits for vibration comfort are summarized in Table 1, which correspond to three vibration-related factors, respectively natural frequency, peak acceleration and RMS-based response factor.

**Table 1.** The international standards and limits for vibration comfort.

| Standards Name | Assessment Criteria | Category | Limitations |
|---|---|---|---|
| GB50010-2010 [19] (Chinese code) | Natural frequency | Residential and Apartment | f ≥ 5 Hz |
| | | Office and Hotel | f ≥ 4 Hz |
| | | Public buildings | f ≥ 3 Hz |
| IS02631-2 [20] (International) | Peak acceleration limit | Office and residences | $10\times$ baseline $a_{rms}$ |
| | | Indoor footbridges, shopping mall | $28\times$ baseline $a_{rms}$ |
| | | Rhythmic activities | $100\times$ baseline $a_{rms}$ |
| SCI [21] (European code) | Response factor R | Office | 8–10 |
| | | Residential | 4–8 |
| | | Operating room | 1–4 |

### 1.1.5. Finite Element Studies of Vibration Comfort

At present, the international research on human-induced vibration comfort of the building floor is mainly based on the experimental method and the finite element method. With the improvement of finite element techniques in recent years, the use of finite element analysis methods to simulate floor slab vibration activity has become a common research method. Eriksson [22] have carried out extensive experimental studies and numerical analyses of simply supported prestressed floor covers, and found that for floor slab systems with characteristic frequencies below 25 Hz, the dynamic modulus of elasticity of prestressed concrete can be obtained by fitting the finite element model to the first vibration mode. El-Dardiry, Wahyuni [23] have determined that the slab-column model simulates a structure that is more in accordance with the actual structure by comparing several floor finite element models. The plate and column model are able to accurately simulate and measure the inherent frequencies of the finite element model. Therefore, in this paper the slab and column model will be used to construct the finite element model.

## 2. Theory of the Post-Tensioned Flat Slab

Prestressed concrete, as a special form of reinforced concrete, is designed to reduce or counteract the internal pressure by applying compressive loads to the concrete structure during construction [24]. Based on the time of application of prestress, prestressed concrete is divided into pretensioned concrete and post tensioned concrete. In contrast, post tensioned concrete is used in most construction projects, as it uses portable hydraulic instruments to apply tension without the need for a pedestal [25]. Compared with the beam-slab floor system, the post tensioned flat slab without beams is particularly attractive in high-rise buildings because of its lower total floor height with smooth and free space under the slab [26].

### 2.1. Two Types of Post Tensioned Concrete Slab

There are two types of post tensioned concrete slab, depending on the contact between the prestressing tendons and the concrete.

Bonded prestressed concrete: Prestressing tendons in pipes can be bonded to the concrete contact surface by use of a pressure grouting to form bonded prestressed concrete. Because the tendon cannot move freely, so the prestressed tendon and the concrete around the bond have the same strain change. The bond action enables the steel tendon to produce sufficient force in a short span [25].

Unbonded prestressed concrete: Restressing tendons can be moved in the pipe without bonding to the concrete contact surface, which is called unbonded concrete. Since the

tendons can slide in the concrete, the tendons are constantly adjusted between adjacent anchors and tend to be uniformly distributed [27].

According to the Australian Standard AS3600-2018 [28] Concrete Construction Clause 17.3.5: unbonded prestressed concrete can only be applied in slabs at ground level, which cannot be used as floor slabs in high-rise buildings, therefore this paper will concentrate on the property of bonded post-tensioned flat slab.

### 2.2. Bonded Post Tension Flat Slab Design Method

The design of the bonded post-tensioned concrete slabs was carried out using the equivalent frame method, the specific design process of which is referred to in Naaman [29] design guide as follows:

a. Design slab thickness ratios based on the span-to-depth ratio and fire resistance requirements;
b. Calculate the prestress values required to balance the equivalent frame loads using the load balancing method;
c. Calculate prestressing tendons area and layout to check that spacing requirements are met;
d. Superimpose prestressing stresses and unbalanced stresses and compare superimposed stresses with allowable stresses
e. Check the strength limit states (ultimate flexural strength, shear strength)
f. Check the serviceability limit states (deflection, vibration)

#### 2.2.1. Initial Analysis

The thickness of the slab must provide sufficient stiffness to ensure serviceability limits for deflection, vibration and fire protection. the Post-Tensioning Institute [30] has given experimental recommendations for the design of tensioned slab thickness based on span length, as shown in the Table 2. For flat slab, a span-depth ratio of 45 will be used for design purposes.

**Table 2.** The span-to-depth ratios limitation.

| Floor System Type | Span-to-Depth Ratio (L/D) |
|---|---|
| Flat plate | 45 |
| One-way slab | 48 |
| Flat slab with drop panels | 50 |
| Edge-supported slab | 55 |
| Waffle slab | 35 |

#### 2.2.2. Prestressing Force

Clause 17.3.4.6 of AS3600-2018 [28]: Concrete structures gives the limitations of the maximum jacking force applied to the post-tensioned tendons, $P_{jack-max}$, as shown in the Equation (3), where $f_{pb}$ is characteristic minimum breaking strength, and $A_p$ is cross-sectional area of prestressed tendons:

$$P_{jack-max} = 0.85 f_{pb} A_p \tag{3}$$

Losses of jacking force occur during construction for various reasons. Losses in prestressing are composed of direct losses and time-dependent losses, the loss percentage depending to some extent on the method and equipment used for prestressing as well as on the shrinkage and creep variation of the concrete [31]. AS3600-2018 Section 3.4 gives details of the calculation of prestressing losses for different conditions. For general low relaxation tendons, a total loss of approximately 20% of the jacking force is desirable for safety considerations [32]. The equation $P_e$ for the effective prestress is shown in Equation (4):

$$P_e = 0.80 \times P_{jack-max} \tag{4}$$

Due to the relatively high forces applied to the prestressed concrete section, the load-bearing stresses in the anchorage zone of the post-tensioned member are significantly larger. In order to ensure security of the structure, a characteristic strength in the range of 40 to 65 MPa for post tension slab is required. Moreover, higher concrete strengths are more conducive to the transfer of prestress in the bonded reinforcement bundles [31].

### 2.2.3. Load Balance Approach

Load balancing is a method of finding the appropriate prestress and tendons profile with a specific loading [31]. The preload for a partial tendon profile applied to the concrete is described as the equivalent force, which is shown in Figure 1:

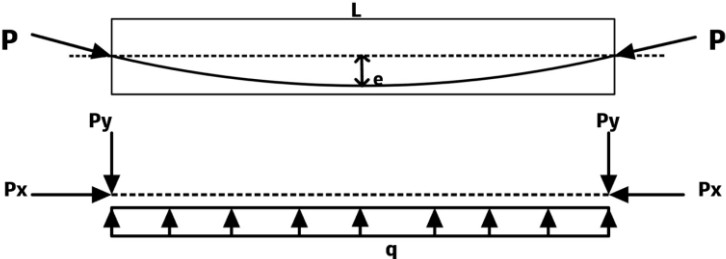

**Figure 1.** Equivalent forces of partial tendon profile.

The required effective prestress *P* for a balanced uniformly distributed dead load *w* in the drape *e* parabolic tendon is calculated by the Equation (5):

$$P = \frac{wL^2}{8e} \tag{5}$$

Load balancing on the flat plate is achieved by laying prestressing tendons to counteract the dead load. The prestressing tendons are distributed in the long span and short span directions of the slab plane and work together to provide the required load twice as much as expected. For slab a balance of 65% to 80% of the constant dead load is generally required. For two-way slabs using more than four flat conduit tendons, the prestress level in each direction is typically in the range P/A = 1.2–2.6 MPa [31].

### 2.2.4. Tendons Layout

Prestressed steel in Australia must conform to the requirements of AS/NZS4672.1 There are various methods of laying tendons, but as the edges of flat slab column supports are usually insufficiently high strength, the preferred tendon layout for two-way slabs is to concentrate tendons adjacent to the edges of the column supports in the long span direction (column line tendons) and distribute uniformly in the other direction (slab tendons) [33]. The tendon spacing is not restricted in AS3600-2018, but ACI-318-11M requires that for uniformly distributed loads the tendon spacing in at least one direction should not exceed 8 times the thickness of the slab. The ACI restriction recommendations will be used in this paper [31].

### 2.2.5. The Strength and Serviceability Limit States

For the designed bonded post-tension concrete flat slab, it needs to be checked for strength and serviceability limit states and the checking methods and standards will be carried out in accordance with AS3600-2018. The specific tests are shown in the Table 3.

**Table 3.** The strength and serviceability limit states requirements.

| Check Limit States | Limit Value | Code |
|---|---|---|
| Average prestress P/A | 1.2–2.6 MPa | Gilbert et al. [31]. |
| Ultimate strength in bending | $M^* \le \phi 1.2[Z\left(f'_{ct.f} + \frac{P_e}{A_g}\right) + P_e e)]$ | Clause 8.1.6.1 |
| Maximum Compressive Stress | $0.45 f'_c$ | Clause 18.1 |
| Shear stress | $V^* \le \frac{\phi * V_{uo}}{\left[1.0 + \frac{u M^*_v}{(8V^* a d_{om})}\right]}$ | Clause 9.3.4 |
| Deflection | l/250 | AS3600-2018 Table 2.3.2 |

*2.3. Summary*

This section introduces the basic concepts and classification of post-tensioned concrete, and gives the design process for the most commonly used bonded post-tensioned concrete flat slab in Australia. For each step of the design process, this section provides theoretical concepts and references. In this paper, to simplify the calculations, prestressing tendons will be considered to provide sufficient strength in replacement of the normal reinforcement bars. In addition, prestressing tendons anchorage devices and time-based prestress losses will be ignored in the numerical analysis, and their implications to natural frequency will need to be investigated separately and in depth in the future. This paper will follow the above design theories and guidelines to design bonded post-tensioned concrete flat slab models with different geometries and structural properties, and investigate the factors influencing vibration serviceability.

**3. Methodology**

In this investigation, a structural design of a 10 m × 9 m bonded post-tensioned concrete flat slab is used as a reference model, where the concrete slab is supported at each corner by a 400 × 400 mm column. The combination of four flat slabs is used to simulate the effect of real structural changes and adjacent slabs in a high-rise building. The structural factors affecting the free and forced vibrations of the post-tensioned concrete slab are explored by varying the length, width, thickness and damping ratio of the post tensioned floor system. The material properties were maintained consistent for all models, and the results were compared by the controlled variable method.

In particular, the natural frequency f, the peak acceleration of pedestrian forced vibration and response factor R can be calculated using the FEA method in Strand7 and SCI and CSTR43 standard guideline method. The natural frequency, the forced vibration peak acceleration and response factor R are used as evaluation criteria.

*3.1. Model Design*

The design of a post-bonded tension concrete flat slabs are approached through the floor system design process in Theory, with reference to AS360-2018, AS/NZS4672.1, ACI-318-11M standards. For the properties of concrete, the plastic failure behaviour and tension damage of concrete were considered in the design. The concrete properties will be determined in AS3600-2018. The material properties of the specified strengths of concrete are shown in the following Table 4.

**Table 4.** The concrete material properties used in models.

| Concrete Grade | Modules (MPa) | Density (kg/m³) | Poisson' Ratio |
|---|---|---|---|
| 40 | 32,800 | 2400.0 | 0.2 |

Prestressing steel tendons: 12.7 mm diameter 7-wires low relaxation steel manufactured by Australian Company Fortec Australia Pty Ltd [34]. in accordance with AS/NZS4672.1, which is the type of prestressing steel commonly used in Australia. Using

the FMA505 system, 5 No. 12.7 mm diameter strands placed inside a 70 mm flat duct. The material properties and proof force and layout limitations of the prestressing tendons FMA505 systems are given in the Table 5.

**Table 5.** The properties and layout limitations of the FMA505 tendons [34].

| | No. Strands | 5 No. 12.7 mm Strands |
| --- | --- | --- |
| | Modules (MPa) | 200,000 |
| | Nominal tensile strength (MPa) | 1870 |
| FMA505 | Breaking Load (kN) | 920 |
| | Proof Force (kN) | 780 |
| | Minimum Concrete Thickness (mm) | 140 |
| | Minimum Anchor Spacing (mm) | 300 |

The thickness of the reference model of the post-tensioned concrete slab is designed to be 270 mm. The concrete slab will be supported by 400 mm × 400 mm columns at each corner, the connections are hinged. The ideal type of parabolic shape of the prestressing steel tendons in concrete is shown in the Figure 2. Balancing 80% of the dead load by the load balance method requires the layout of 10 prestressing tendons in the long span direction and 8 prestressing tendons in the short span direction for each flat slab. The specific layout and spacing meet the criteria of ACI-318-11M and the FMA505 system as shown in the Figure 3.

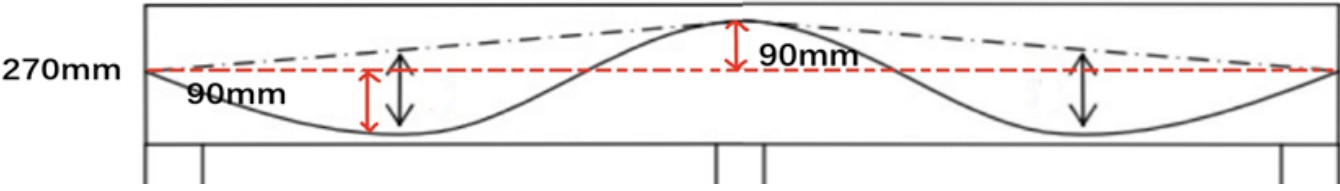

**Figure 2.** The ideal type of parabolic shape of the prestressing tendons for reference model.

The models created are divided into 4 categories according to the geometric and performance parameters to be studied, the parameters of the specific models are summarized in the Table 6. The models and tendons will be designed in the same way as above, and for effective comparison, the same number and depth of prestressing tendons will be used.

### 3.2. Strand7 Finite Element Modelling and Analysis Process
### 3.2.1. The Modelling Process of Strand7

The concrete slab was created using Quad4 cells with specific dimensions and material properties designed in Table 6. The prestressed tendons were simplified to a rectangle of 63.5 mm × 12.7 mm. Ideal parabolic curves were constructed using Beam2 cells to assign material properties. As the prestressing tendons and concrete of the bonded post-tensioned concrete slab are connected by bond, the connection needs to be made using the Strand7 link connection function package. Firstly, a high modulus pie-shaped attachment beam was added to the points on the prestressing tendon, then the attachment path was used to establish the connection between the prestressing tendons and the concrete internal horizontal plane. Finally, the attachment beam is converted into a rigid link using the convert function to connect the prestressing tendons to the concrete in the x-y-z plane.

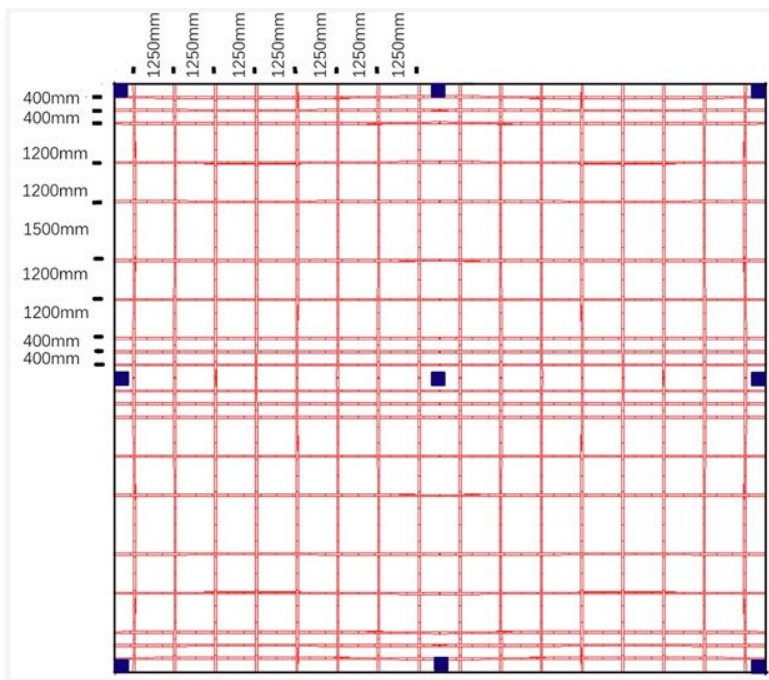

**Figure 3.** The specific prestressing tendons layout for reference model.

**Table 6.** The geometric and performance parameters of each model.

|  | Length (m) | Width (m) | Slab Thickness (mm) | Damping Ratio | Concrete Characteristic Strength (MPa) |
|---|---|---|---|---|---|
| Reference Model | 10 | 9 | 270 | 0.02 | 40 |
| Model 2.1 | 12 | 9 | 270 | 0.02 | 40 |
| Model 2.2 | 8 | 9 | 270 | 0.02 | 40 |
| Model 3.1 | 10 | 11 | 270 | 0.02 | 40 |
| Model 3.2 | 10 | 8 | 270 | 0.02 | 40 |
| Model 4.1 | 10 | 9 | 300 | 0.02 | 40 |
| Model 4.2 | 10 | 9 | 330 | 0.02 | 40 |
| Model 5.1 | 10 | 9 | 270 | 0.03 | 40 |
| Model 5.2 | 10 | 9 | 270 | 0.04 | 40 |

After the connection, the preload was applied to the prestressing tendons using the preload function. The connection between the concrete slab and the column was made by means of a common point, which was a hinged connection. The connection point was free to rotate and the connection point was designed to be 200 mm from the edge of the concrete slab to ensure that the column fully supports the concrete slab. The Figure 4 shows a dimetric view of the reference model in Strand7. In order to simulate the dynamic behaviour caused by pedestrian loads, an equivalent force of 746 N was applied at the midpoint of the concrete slab.

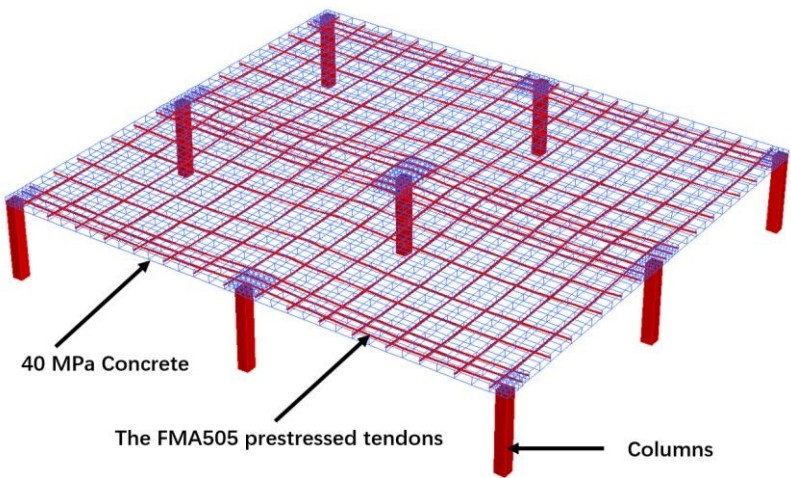

**Figure 4.** The dimetric view of the reference model in Strand7.

3.2.2. The Analysis Processes of Strand7

After constructing the models, free vibrations and forced vibrations are analysed using the Strand7 function package. The natural frequency and vibration modes of the model were calculated using the natural frequency function within solver. According to the Strand 7 Low Frequency Vibration Sensitive Structures guide [35], 50 vibration structures for pre-stressed slabs were analysed. The natural frequency is the first vibration mode frequency of the models. For vibrations caused by walking forced loads, a footprint analysis method using the Strand7 Harmonic Response Solver. Footprint-induced vibration of structures is assessed using walking load frequencies. Pedestrian walking in the field can be effectively simulated by applying a hypothetical load point chosen to excite the shape of the critical mode. The load application point is chosen at the point of maximum deflection of the slab to obtain the maximum dynamic response.

Based on literature review and Strand7 guide, for pedestrian load frequencies the first four Fourier harmonic functions will be used. The relationships in the Table 7 are constructed in the Factor vs. Frequency Table in Strand7, and the appropriate DLF equation is selected to analyses the peak acceleration due to each of the first four harmonics separately using the harmonic frequency solver.

**Table 7.** The relationship between dynamic load factor and frequency.

| N.O. Fourier Harmonic Function | Frequency (Hz) | Design Dynamic Load Factor |
|---|---|---|
| 1st harmonic Design dynamic load factor | 1.0<br>2.316<br>2.8 | 0.0205<br>0.56<br>0.56 |
| 2nd harmonic Design dynamic load factor | 2.0<br>5.6 | 0.0802<br>0.10036 |
| 3rd harmonic Design dynamic load factor | 3.0<br>8.4 | 0.0522<br>0.08676 |
| 4th harmonic Design dynamic load factor | 4.0<br>11.2 | 0.039<br>0.0858 |

The following three values were calculated using the harmonic analysis of Strand7: Peak Acceleration $a_{peak}$ and Response factor R.

The peak acceleration $a_{peak}$ is calculated from the linear summation of the amplitudes of the individual components:

$$Total\ a_{peak} = \sum_{Harmonics} a_{h,peak} \tag{6}$$

where:

$a_{h,peak}$ = the amplitudes of the individual harmonic components

The response factor R is obtained as the square root of the sum of the squares of each harmonic baseline response factor. The baseline response factor is given in the Equation in several different methods depending on the frequency.

$$R_{base} = \sqrt{2}\frac{1}{100\sqrt{f}} \qquad f < 4Hz \tag{7}$$

$$R_{base} = \sqrt{2}\frac{1}{200} \qquad 4Hz \leq f \leq 8Hz \tag{8}$$

$$R_{base} = \sqrt{2}\frac{1}{200}\frac{f}{8} \qquad 8Hz < f \tag{9}$$

$$R = \sqrt{R_1^2 + R_2^2 + R_3^2 + R_4^2} \tag{10}$$

where:

$f$ = Frequency values from each harmonic response solver.

### 3.3. Simplified Hand Calculation of the Critical Dynamic Parameters

3.3.1. Natural Frequency

The SCI standard focuses on the calculation of the vibration performance of lightweight concrete composite slabs supported by steel beams. Although most of the calculation theory involved is common to other forms of floor cover systems, direct use would result in errors. CSTR43 gives the calculation of the two-way post-tensioned concrete flat slab based on SCI for natural frequency of composite floors [36]. This standard assumes that the floor slab is simply supported on all four sides. The specific calculation equation is shown in Equation (11) [37].

$$f = \frac{\pi}{2}\sqrt{\frac{n_a^4}{(n_x l_x)^4}EI_x + \frac{n_b^4}{(n_y l_y)^4}EI_y} \times \sqrt{\frac{1}{m}} \tag{11}$$

where:

$n_x l_x$ = side length in the x-direction of the rectangular slab;
$n_y l_y$ = side length in the y-direction of the rectangular slab;
$n_a$ = the number of half waves in the x-direction of the rectangular slab;
$n_b$ = the number of half waves in the y-direction of the rectangular slab;
$EI_x$ = flexural stiffness in the x-direction of the rectangular slab;
$EI_y$ = flexural stiffness in the x-direction of the rectangular slab;
$m$ = mass of building cover per unit area.

3.3.2. Peak Acceleration

The SCI and AISC/CISC standards recommend that for continuous steady state response, the peak acceleration $a_{peak}$ can be calculated using Equation (12) [3].

$$a_{peak} = \frac{\alpha_n P_0}{M}\frac{1}{2\xi} \tag{12}$$

where:

$\alpha_n$ = the Fourier coefficient of the ith harmonic;
$P_0$ = hydrostatic force applied by a person (Assumed taken as 746 N);

$M$ = the modal mass;

$\xi$ = damping ratio.

The Fourier coefficients can be evaluated based on the curves concluded from the experimental study by Rainer et al. [10]. The Table 8 shows the average values of the Fourier coefficients for the walking activity in conclusion by Rainer et al. [10].

**Table 8.** The average values of the Fourier coefficients for the walking activity.

| N.O. | First Harmonic $\alpha_1$ | Second Harmonic $\alpha_2$ | Third Harmonic $\alpha_3$ | Fourth Harmonic $\alpha_4$ |
|---|---|---|---|---|
| Average $\alpha_n$ | 0.33 | 0.14 | 0.05 | 0.05 |

### 3.3.3. Response Factor

The calculated Response factor R enables the forced vibration performance of the slab to be evaluated by comparing it to standard specifications. According to the guidance in SCI P354 Section 7.5 [21], after finding the natural frequency of the post-tensioned concrete slab, the dynamic response can be calculated using the Equation (13).

$$esponse\ factor\ R = \frac{a_{rms}}{0.005} \tag{13}$$

where:

$a_{rms}$ = the root mean square (RMS) acceleration of slab.

SCI divides floors into high frequency floors and low frequency floors according to the fundamental frequency, and there are different methods for calculating the RMS acceleration for the two types of floors

For the natural frequencies between 3 Hz and 10 Hz (low frequency floors), the RMS acceleration is calculated by Equation (14).

$$a_{rms} = \mu_e \mu_r \frac{0.1P_0}{2\sqrt{2}M\xi}\ W\rho \tag{14}$$

For the natural frequency more than 10 Hz (high frequency floor), the RMS acceleration is calculated by Equation (15).

$$a_{rms} = 2\mu_e \mu_r \frac{185}{Mf_0^{0.3}}\ \frac{P_0}{700}\frac{1}{\sqrt{2}}W \tag{15}$$

where:

$\mu_e$ = modal shape factor of the excitation point, assumed to be 1 without shape factor;

$\mu'_r$ = modal shape factor at the reaction point, assumed to be 1 without shape factor;

$P_0$ = hydrostatic force applied by a person (Assumed taken as 746 N);

$\xi$ = damping ratio;

$M$ = the modal mass (kg);

$\rho$ = resonance build-up factor;

$f_0$ = fundamental frequency;

$W$ = appropriate canonically defined weighting factor for human vibration perception based on fundamental frequency.

### 3.4. Comparison of FEA Results with Standard Calculation Results

The FEA model results obtained in Section 4.2. and the standard calculation results in Section 4.3. will be collected in Excel for analysis. The effect of the post-tensioned concrete slab model length, width, thickness and damping ratio on the three vibration serviceability parameters is investigated by means of graphical representations. In the FEA, multiple subdivision dimensions are not used to account for the effects of mesh sensitivity and

there are errors in the data from the FEA model. The specific results of the comparison are presented in the next section.

*3.5. Methodology Limitation*

This method has some limitations due to the simplification of some structural details in the modelling process of the finite element analysis. The prestressed tendons in the finite element model are assumed to be long rectangular bars and placed in the concrete according to an ideal parabolic shape, the structural properties obtained may be subject to some error. In following the standard calculations, the model is simplified to four points as simple supports, neglecting the effect of column supports, which may have some effect on the calculation of natural frequency. Although there are some limitations, the Strand7 finite element models are analysed in accordance with the official guidance documents, the results obtained are generally reasonable. A comparative analysis of the errors in the two approaches will be given below section.

**4. Comparison of Results**

The results will be collected and processed as described as Section 3.4. This results Section 4 is divided into four parts. The first three parts correspond to the effects of the different structural dimensions and damping ratios for the two methods on the natural frequency, peak acceleration and response factor, in order to determine the effect of structural dimensions and damping ratios on the vibration serviceability of post-tensioned floor systems. The fourth parts compare the error ratios of the two methods for the same model to determine the reasonableness and practicality of the finite element analysis method.

*4.1. The Effect of the Structural Dimensions and Damping Ratios on Natural Frequency*
4.1.1. The Effect of the Long-Span Length on Natural Frequency

The natural frequency results corresponding to different long-span lengths in category 1 are shown in Table 9. Figure 5 demonstrates the decreasing trend of the natural frequency results with increasing Long-Span length in the Strand7 FEA results. the CSTR43 results show a similar trend. The results of the two methods are in high concordance, but the CSTR43 results are progressively larger than the FEA results as the length continues to increase.

**Table 9.** The results of natural frequency calculated by change the long-span lengths.

| | Long-Span Length (m) | CSTR43 Solution (Hz) | Strand7 FEA Solution (Hz) |
|---|---|---|---|
| Reference Model | 10 | 5.5078 | 5.9886 |
| Model 2.1 | 12 | 4.9947 | 4.3577 |
| Model 2.2 | 8 | 6.5213 | 7.2540 |

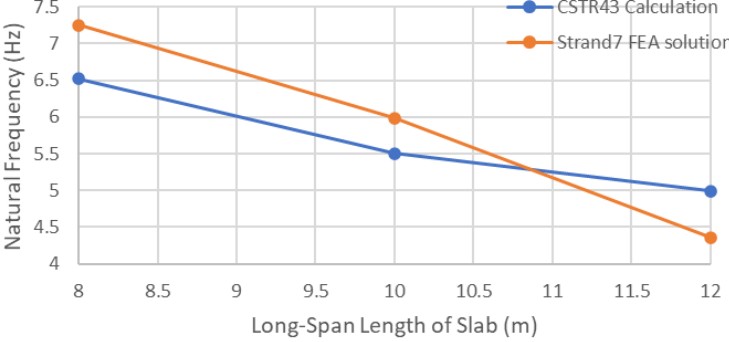

**Figure 5.** The Effect of the Long-Span Length on Natural Frequency.

### 4.1.2. The Effect of the Short-Span Length on Natural Frequency

The results for the natural frequency corresponding to different short-span lengths (width) in the category 2 are shown in Table 10. It is evident from Figure 6 that both CSTR43 calculation and Strand7 FEA solution show a decreasing trend as the short-span length increases. Therefore, the natural frequency and width in the post-tensioned floor show a negative correlation. In addition, the results obtained by Strand7 are greater than those obtained by CSTR43 results.

**Table 10.** The results of natural frequency calculated by change the short-span lengths.

|  | Short-Span Length (m) | CSTR43 Solution (Hz) | Strand7 FEA Solution (Hz) |
|---|---|---|---|
| Reference Model | 9 | 5.5078 | 5.9886 |
| Model 3.1 | 11 | 4.7329 | 5.2034 |
| Model 3.2 | 8 | 6.1459 | 6.2869 |

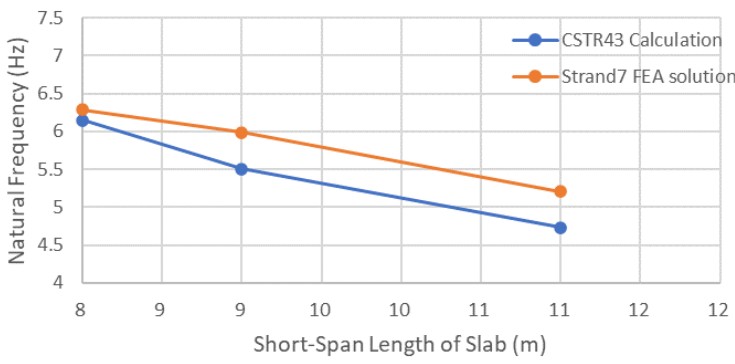

**Figure 6.** The Effect of the Short-Span Length on Natural Frequency.

### 4.1.3. The Effect of the Slab Thickness on Natural Frequency

The results for the natural frequency corresponding to different slab thicknesses in the category 3 model are shown in Table 11. In Figure 7 the results for both CSTR43 and strand7 show that the natural frequency get larger as the slab thickness increases. The trends of the two results are relatively similar, but the results obtained for Strand7 are greater than the results calculated for CSTR43.

**Table 11.** The results of natural frequency calculated by change the slab thickness.

|  | Slab Thickness (m) | CSTR43 Solution (Hz) | Strand7 FEA Solution (Hz) |
|---|---|---|---|
| Reference Model | 0.270 | 5.5078 | 5.9886 |
| Model 4.1 | 0.300 | 6.1198 | 6.6534 |
| Model 4.2 | 0.330 | 6.7318 | 7.3182 |

### 4.1.4. The Effect of the Damping Ratio on Natural Frequency

The results for the natural frequency corresponding to different damping ratios in the category 4 model are shown in Table 12. Figure 8 shows that the natural frequency does not change with increasing damping ratio in the strand7 FEA results, and the CSTR43 calculations show the same relationship. Therefore, the change in damping ratio has no effect on the natural frequency. In addition, the Strand7 results are larger than the CSTR43 calculated results.

**Effect of Slab Thickness on Natural Frequency**

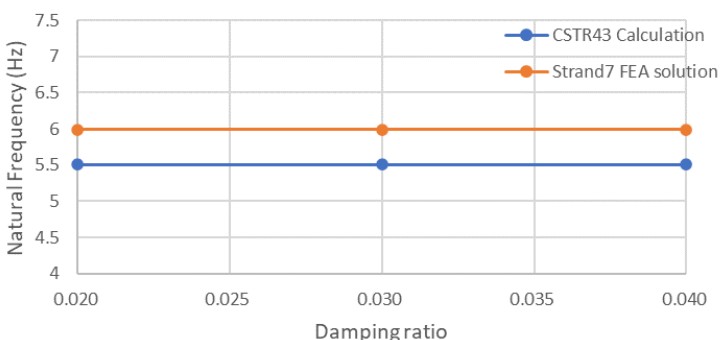

**Figure 7.** The Effect of Slab Thickness on Natural Frequency.

**Table 12.** The results of natural frequency calculated by change damping ratio.

|  | Damping Ratio | CSTR43 Calculation (Hz) | Strand7 FEA Solution (Hz) |
| --- | --- | --- | --- |
| Reference Model | 0.02 | 5.5078 | 5.9886 |
| Model 5.1 | 0.03 | 5.5078 | 5.9886 |
| Model 5.2 | 0.04 | 5.5078 | 5.9886 |

**Effect of Slab Damping Ratio on Natural Frequency**

**Figure 8.** The Effect of Slab Damping Ratio on Natural Frequency.

*4.2. The Effect of the Structural Dimensions and Damping Ratios on Peak Acceleration*

4.2.1. The Effect of the Long-Span Length on Peak Acceleration

The results for different long-span lengths in the category 1 model corresponding to peak acceleration are shown in Table 13. In Figure 9, the results for SCI and Strand7 both show that the peak accelerations tend to decrease as the long-span length increases, and the trends are quite similar for both results. In addition, the results obtained for Strand7 are less than SCI Approach results.

**Table 13.** The results of peak acceleration calculated by change long-span length.

|  | Long-Span Length (m) | The SCI Approach(%g) | Strand7 FEA Solution (%g) |
| --- | --- | --- | --- |
| Reference Model | 10 | 0.420 | 0.399 |
| Model 2.1 | 12 | 0.345 | 0.280 |
| Model 2.2 | 8 | 0.536 | 0.508 |

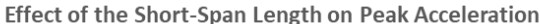

**Figure 9.** The Effect of the Long-Span Length on Peak Acceleration.

### 4.2.2. The Effect of the Short-Span Length on Peak Acceleration

The results for different short-span lengths in the category 2 model corresponding to peak acceleration are shown in Table 14. The SCI and strand7 results in Figure 10 shows that peak acceleration and short-span length (width) show a negative correlation, with peak acceleration gradually decreasing at increasing short-span length. The SCI results are larger than Strand7 FEA solution at initial stage, but as the width increases, the two results tend to be similar.

**Table 14.** The results of peak acceleration calculated by change short-span length.

|  | Short-Span Length (m) | The SCI Approach (%g) | Strand7 FEA Solution (%g) |
|---|---|---|---|
| Reference Model | 9 | 0.420 | 0.399 |
| Model 3.1 | 11 | 0.337 | 0.341 |
| Model 3.2 | 8 | 0.478 | 0.428 |

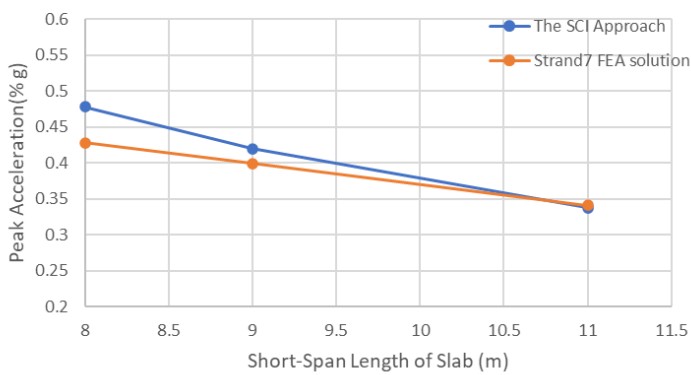

**Figure 10.** The Effect of the Short-Span Length on Peak Acceleration.

### 4.2.3. The Effect of the Slab Thickness on Peak Acceleration

The results for different slab thicknesses in the category 3 model corresponding to the peak acceleration are shown in Table 15. Figure 11 shows that for the Strand7 FEA results, the peak acceleration decreases as the slab thickness increases, but the change is slight. The results for the SCI approach show the same trend. The results for the two approaches are not very significantly different, but the tendency for the Strand7 results to vary less than the SCI results.

**Table 15.** The results of peak acceleration calculated by change slab thickness.

|  | Slab Thickness (m) | The SCI Approach (%g) | Strand7 FEA Solution (%g) |
|---|---|---|---|
| Reference Model | 0.27 | 0.420 | 0.399 |
| Model 4.1 | 0.3 | 0.378 | 0.380 |
| Model 4.2 | 0.33 | 0.343 | 0.363 |

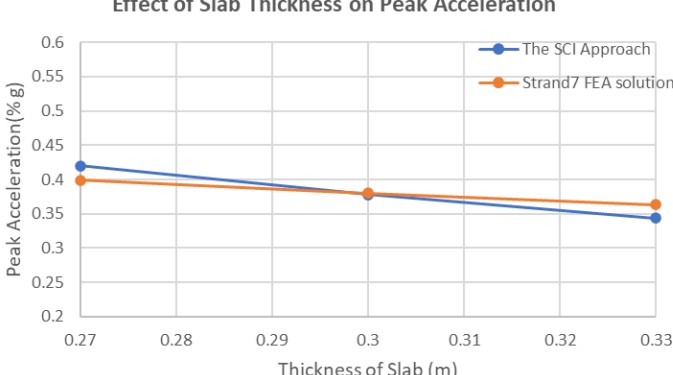

**Figure 11.** The Effect of Slab Thickness on Peak Acceleration.

### 4.2.4. The Effect of the Damping Ratio on Peak Acceleration

The results for the peak acceleration corresponding to different damping ratios in the category 4 model are shown in Table 16. Figure 12 shows that for the strand7 FEA results, the peak acceleration tends to decrease as the damping ratio increases and the trend becomes slower as the damping ratio increases. the SCI shows the same trend and the difference in values between the two methods is slight, but the SCI changes to a greater extent than the Strand7 FEA result.

**Table 16.** The results of peak acceleration calculated by change damping ratio.

|  | Damping Ratio | The SCI Approach (%g) | Strand7 FEA Solution (%g) |
|---|---|---|---|
| Reference Model | 0.02 | 0.420 | 0.399 |
| Model 5.1 | 0.03 | 0.280 | 0.325 |
| Model 5.2 | 0.04 | 0.210 | 0.278 |

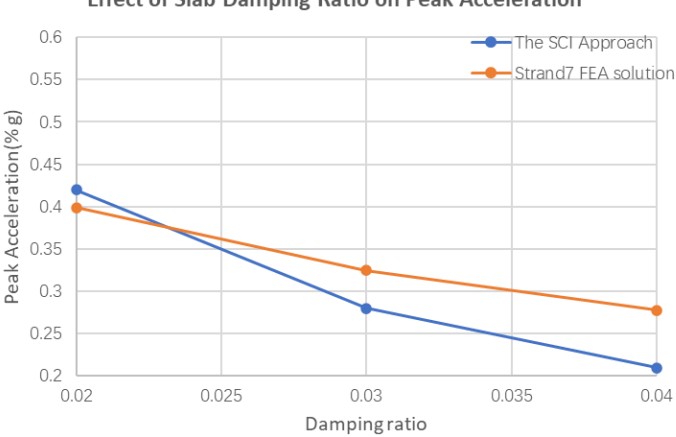

**Figure 12.** The Effect of Slab Damping Ratio on Peak Acceleration.

### 4.3. The Effect of the Structural Dimensions and Damping Ratios on Response Factor R

4.3.1. The Effect of the Long-Span Length on Response Factor R

The results for the response factor R corresponding to different long-span lengths in the category 1 model are shown in Table 17. Figure 13 shows that for both the Strand7 FEA and SCI results, as long-span length increases, the response factor shows a decreasing trend. The difference between the two methods is not large, but the decreasing trend shows a different pattern: with increasing long-span length, SCI decreasing trend slows down, the opposite trend for Strand7.

**Table 17.** The results of response factor R calculated by long-span length.

|  | Long-Span Length (m) | The SCI Approach | Strand7 FEA Solution |
|---|---|---|---|
| Reference Model | 10 | 4.316 | 4.47 |
| Model 2.1 | 12 | 3.545 | 2.65 |
| Model 2.2 | 8 | 5.515 | 5.78 |

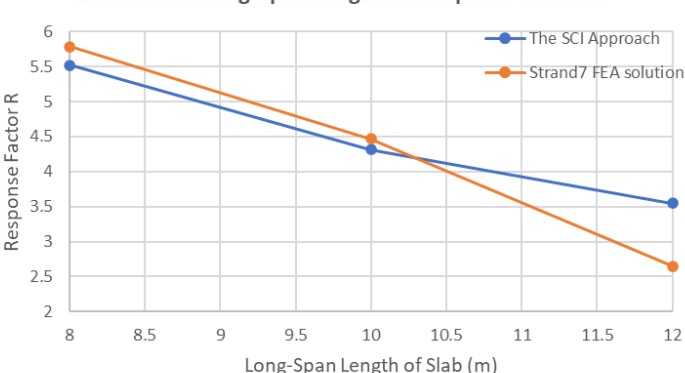

**Figure 13.** The Effect of the Long-Span Length on Response Factor R.

4.3.2. The Effect of the Short-Span Length on Response Factor R

The results for the response factor R corresponding to different short-span lengths in the category 2 model are shown in Table 18. Figure 14 shows that for the Strand7 FEA results, response factor and short-span length show a negative correlation, with increasing short-span length, response factor becoming progressively smaller. SCI approach shows a similar trend and has little difference from Strand7. In addition, the decreasing trend of SCI slowing down as the width gradually increases, the opposite trend for Strand7, which is similar to the relationship between long-span length and response factor.

**Table 18.** The results of response factor R calculated by short-span length.

|  | Short-Span Length (m) | The SCI Approach | Strand7 FEA Solution |
|---|---|---|---|
| Reference Model | 9 | 4.316 | 4.47 |
| Model 3.1 | 11 | 3.470 | 3.59 |
| Model 3.2 | 8 | 4.916 | 4.56 |

**Effect of the Short-Span Length on Response Factor R**

**Figure 14.** The Effect of the Short-Span Length on Response Factor R.

### 4.3.3. The Effect of the Slab Thickness on Response Factor R

The results of the response coefficients R corresponding to different slab thicknesses in the category 3 model are shown in Table 19. Figure 15 shows the results for SCI and Strand 7, where the response factor increases with greater slab thickness, but the magnitude of the increase is smaller. It is implied that there is a negative correlation between response factor and plate thickness, but the effect of plate thickness is slight. In addition, the SCI results are greater than the Strand7 FEA results.

**Table 19.** The results of response factor R calculated by slab thickness.

|  | Slab Thickness (m) | The SCI Approach | Strand7 FEA Solution |
|---|---|---|---|
| Reference Model | 0.27 | 4.316 | 4.47 |
| Model 4.1 | 0.3 | 3.885 | 4.26 |
| Model 4.2 | 0.33 | 3.620 | 3.92 |

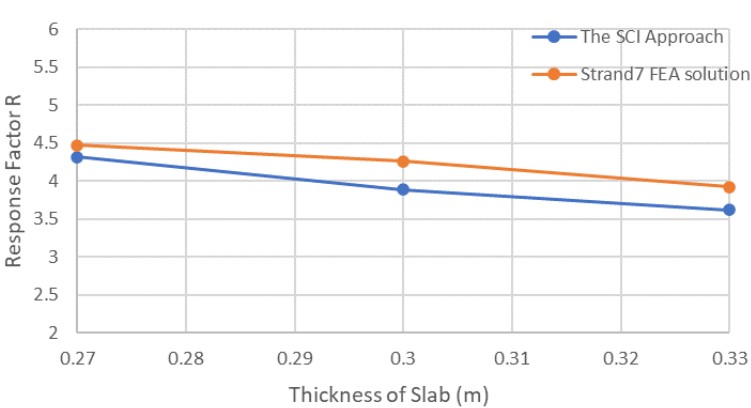

**Figure 15.** The Effect of the slab thickness on Response Factor R.

### 4.3.4. The Effect of the Damping Ratio on Response Factor R

The results of the response factor R for different damping ratios in the category 4 model are shown in Table 20. Figure 16 shows that for the Strand 7 FEA results, the response factor and damping ratio show a negative correlation, and as the damping ratio increases, the response factor decreases at a slower rate. The results of SCI showed a similar trend, with a small difference in values between the two methods.

**Table 20.** The results of response factor R calculated by damping ratio.

|  | Damping Ratio | The SCI Approach | Strand7 FEA Solution |
|---|---|---|---|
| Reference Model | 0.02 | 4.316 | 4.47 |
| Model 5.1 | 0.03 | 2.949 | 3.44 |
| Model 5.2 | 0.04 | 2.212 | 2.79 |

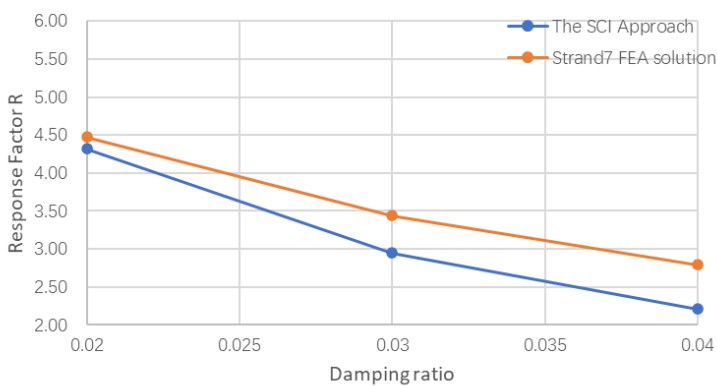

**Figure 16.** The Effect of the damping ratio on Response Factor R.

*4.4. Error Analysis between Standards Calculation Results and Strand7 FEA Solution*

4.4.1. Error Analysis for Natural Frequency

The results of the CSTR43 and SCI criterion are considered to be the more accurate values. By comparing the Strand7 FEA results with CSTR43 results, the accuracy and reasonableness of the FEA model results can be assessed. The natural frequency calculated errors for each model Strand7 and CSTR43 criteria are given in Table 21, and the average errors are calculated for varying different structural parameters. The maximum error is 12.75% when the long span length is equal to 12. The largest mean error was found for the effect on natural frequency when changing the long span length, at 10.91%. The errors in the strand 7 FEA natural frequency results were all less than 15%, which is considered to be within a reasonable margin of error. This is considered to be within a reasonable margin of error. Therefore, the method of calculating natural frequency by FEA using Strand 7 is feasible.

4.4.2. Error Analysis for Peak Acceleration

Table 22 gives the calculated errors of the peak acceleration for each model Strand7 and SCI criteria, and calculates the average error for varying different structural parameters. The largest error occurred at a damping ratio of 0.04 with a maximum error of 32.49%. The maximum mean error of 17.86% for the peak acceleration finite element results occurred for the effect of changing the damping ratio on the peak acceleration. The results for model 5.2 were larger and exceeded the desired error. When calculating peak acceleration using Strand7 FEA, attention needs to be paid to the effect of damping ratio.

**Table 21.** Possible error of the natural frequency in Strand7 compared to CSTR43 results.

|  | Long-span length (m) | Strand7 error ratio | Average error ratio |
|---|---|---|---|
| Reference Model | 10 | 8.73% | |
| Model 2.1 | 12 | 12.75% | 10.91% |
| Model 2.2 | 8 | 11.24% | |
|  | Short-span length (m) | Strand7 error ratio | Average error ratio |
| Reference Model | 8 | 2.29% | |
| Model 3.1 | 9 | 8.73% | 6.99% |
| Model 3.2 | 11 | 9.94% | |
|  | Slab thickness (m) | Strand7 error ratio | Average error ratio |
| Reference Model | 0.27 | 8.73% | |
| Model 4.1 | 0.30 | 8.72% | 8.72% |
| Model 4.2 | 0.33 | 8.71% | |
|  | Damping ratio | Strand7 error ratio | Average error ratio |
| Reference Model | 0.02 | 8.73% | |
| Model 5.1 | 0.03 | 8.73% | 8.73% |
| Model 5.2 | 0.04 | 8.73% | |

**Table 22.** Possible error of the peak acceleration in Strand7 compared to the SCI results.

|  | Long-span length (m) | Strand7 error ratio | Average error ratio |
|---|---|---|---|
| Reference Model | 10 | 4.92% | |
| Model 2.1 | 12 | 18.77% | 9.65% |
| Model 2.2 | 8 | 5.26% | |
|  | Short-span length (m) | Strand7 error ratio | Average error ratio |
| Reference Model | 9 | 4.92% | |
| Model 3.1 | 11 | 1.08% | 5.48% |
| Model 3.2 | 8 | 10.45% | |
|  | Slab thickness (m) | Strand7 error ratio | Average error ratio |
| Reference Model | 0.27 | 4.92% | |
| Model 4.1 | 0.30 | 0.61% | 3.75% |
| Model 4.2 | 0.33 | 5.72% | |
|  | Damping ratio | Strand7 error ratio | Average error ratio |
| Reference Model | 0.02 | 4.92% | |
| Model 5.1 | 0.03 | 16.17% | 17.86% |
| Model 5.2 | 0.04 | 32.49% | |

### 4.4.3. Error Analysis for Response Factor R

Table 23 gives the calculated error in response factor R for each model Strand7 and SCI standard, and calculates the average error for varying the different structural parameters. The largest error in calculating the response factor occurred for the damping ratio of 0.04, which was 26.12%, and was significantly higher than the other models. The largest average error occurs when varying the damping ratio on the response coefficient, at 15.44%. Similar to the results of the FEA model for peak acceleration, attention needs to be paid to the errors generated by the damping ratio when calculating the response coefficients. The errors in the other variables are within reasonable limits and the finite element analysis of the response coefficients calculated with Strand7 is of some practicality.

**Table 23.** Possible error of the response factor R in Strand7 compared to the SCI results.

|  | Long-span length (m) | Strand7 error ratio | Average error ratio |
|---|---|---|---|
| Reference Model | 10 | 3.56% | |
| Model 2.1 | 12 | 25.26% | 11.21% |
| Model 2.2 | 8 | 4.80% | |
|  | Short-span length (m) | Strand7 error ratio | Average error ratio |
| Reference Model | 9 | 3.56% | |
| Model 3.1 | 11 | 3.46% | 4.75% |
| Model 3.2 | 8 | 7.24% | |
|  | Slab thickness (m) | Strand7 error ratio | Average error ratio |
| Reference Model | 0.27 | 3.56% | |
| Model 4.1 | 0.30 | 9.66% | 7.17% |
| Model 4.2 | 0.33 | 8.29% | |
|  | Damping ratio | Strand7 error ratio | Average error ratio |
| Reference Model | 0.02 | 3.56% | |
| Model 5.1 | 0.03 | 16.63% | 15.44% |
| Model 5.2 | 0.04 | 26.12% | |

*4.5. Summary*

This section presents a comparative image analysis and error analysis of the results from Strand7 FEA and standard calculations. The image analysis found that the trends of natural frequency, peak acceleration and response factor R obtained by Strand7 FEA were essentially similar to the SCI and CSTR43 results when the size and damping ratio of the post-tensioned slabs are varied. The error analysis found that for the three vibration serviceability parameters, the Strand FEA model produced the largest errors when varying the damping ratio. In the next section, the implications of these results, and the reasons for the errors, are discussed specifically.

**5. Discussion about the Results**

This paper uses the three criteria vibration parameters to assess the effect of length, width, thickness and damping ratio on vibration serviceability. Therefore, in order to clearly discuss the effects of each factor, this section will be divided into four main parts:

- The effect of slab length and width on vibration serviceability;
- The effect of slab thickness on vibration serviceability;
- The effect of slab damping ratio on vibration serviceability;
- Assess the feasibility of Strand7 finite element analysis by using errors.

In each section, the effects of individual variables on natural frequency, peak acceleration and response factor are discussed in detail, as well as the recommendations for the dimensions and damping ratios design of post-tensioned flat slab.

*5.1. The Effect of Slab Length and Width on Vibration Serviceability*

The span essentially changes the stiffness of the slab to affect the natural frequency. Figures 5 and 6 shows that the CSTR43 and Strand7 FEA solutions have negative correlation when changing the length of the long-span and short-spans. It means that increasing the span in either direction will lead to a decrease in the natural frequency. A comparison of the results shows that increasing the same length in both directions, the effect of the long-span on the natural frequency is greater than the effect of the short-span. Therefore, with increasing span, the floor system is more susceptible to free vibration and resonance. As shown in Figures 9 and 10, the SCI and Strand7 FEA peak accelerations reduce with increasing span in either direction. The similar trend is observed in Figures 13 and 14, where the response factor decreases with increasing span in either direction. Thus, the floor span is not susceptible to pedestrian load vibration as it increases.

The reason for these results is related to the modal mass: in both Equations (12) and (14), the peak acceleration and the response factor show a negative relationship with the model mass M. With the span increases, the larger model mass will decrease the peak acceleration and the response factor when the excitation resonance harmonic order is not changed. However, Jiang, Cheng [38] found that as the span continues to increase, the natural frequency will reduce and when it is lower than 2.8 Hz, the first order pedestrian load harmonic vibration will cause resonance. In the literature review, Allen and Pernica [12] have shown that as the harmonic order decreases, the resonance frequency becomes greater. Therefore, increasing the span is not effective in safely reducing the peak acceleration and the response factor. It is necessary to adjust the span to increase the natural frequency to ensure that the limits in the standard are exceeded, with particular attention to the length in the long-span direction.

### 5.2. The Effect of Slab Thickness on Vibration Serviceability

Figure 7 shows that the slab thickness and natural frequency have a positive correlation: when the slab thickness is increased by 60 mm, the natural frequency increases over 20%. In the SCI and Strand7 FEA results of Figure 11, increasing slab thickness leads to a reduction in peak acceleration. A comparable relationship is shown between slab thickness and response factor in Figure 15. Due to the small variation in thickness in the design of this experiment, the magnitude of change in peak acceleration and response factor shown is not significant.

According to Literature Review and SCI guidance, an increase in slab thickness directly leads to higher stiffness, which in turn improves the natural frequency of the slab. Therefore, increasing the slab thickness can effectively increase the natural frequency According to Equation (12), larger slab thickness has the higher model mass and natural frequency, which affects the model vibration frequency excited by the pedestrian load and reduces the peak acceleration. Increasing the slab thickness improves the natural frequency while reducing the peak acceleration and response factor. However, it should be noted that increasing the slab thickness results in an increase in the overall weight of the slab, requiring more prestressed steel to balance the dead load of the system, and the increased weight also increases the requirements for the columns, which requires additional economic consideration. Therefore, a reasonable increase in slab thickness can improve the vibration serviceability.

### 5.3. The Effect of Slab Damping Ratio on Vibration Serviceability

The results from the Figure 8 Strand7 FEA show that the damping ratio does not affect the natural frequency. The Strand7 FEA and SCI results in Figures 12 and 16 show that improving the damping ratio reduces the peak acceleration and response factor.

In the natural frequency solver of Strand7, the effect of damping ratio on the natural frequency is not considered. In the guidelines given by SCI and CSTR43, natural frequency as a physical property of the floor system is mainly influenced by material properties, shape, temperature, etc. The effect of damping ratio on natural frequency is also not considered. Therefore, for slab systems, higher damping ratio does not increase natural frequency. In Equations (12) and (14), the damping ratio shows a negative correlation with peak acceleration and the response factor, because higher damping ratio results in more kinetic energy being dissipated through heat and sound which described in Literature Review. The damping is increased causing a reduction in peak acceleration and response factor.

The purpose of the damping ratio research is twofold: firstly, to determine the effect on experimental results of using an analogous hypothetical damping ratio for the same floor system when researching vibration serviceability. The second is to investigate whether increasing the damping ratio is effective in improving the vibration serviceability. The literature review recalls that the damping ratio of a structure is difficult to determine due to orthogonal anisotropy [15], and damping ratio has a large influence on vibration serviceability, so a conservative estimate of the damping ratio of the floor slab is needed

when studying the vibration serviceability. In addition, increasing the damping ratio can effectively improve the vibration serviceability. The damping of the floor slab can be increased by using a damper column or some other devices such as tuned mass dampers [8].

*5.4. Assess the Feasibility of Strand7 Finite Element Analysis by Using Errors*

The main reasons for the error are the use of post-tensioned slabs supported by columns in the finite element model, where the vibration serviceability is higher than the simple support assumed by the theoretical calculations. Furthermore, the effect of mesh sensitivity has not been considered in the finite element analysis of this paper, and the results would be more accurate if the number of subdivisions of the finite element model were increased. The differences between the Strand7 FEA and standard calculated vibration parameters results for varying the length, width, thickness and damping ratio of the models are not significant, within reasonable ranges overall. Therefore, it is feasible and accurate to use Strand7 for FEA to simulate vibration serviceability.

## 6. Conclusions

With the popularity of large span prestressed building floor slabs, the study of vibration serviceability due to pedestrian loads on prestressed concrete slabs has become one of the current concerns. In order to obtain a more comprehensive comprehension of the factors affecting the vibration serviceability of post-tensioned concrete slabs, this paper uses Strand7 to model the dynamic behaviour of free and pedestrian load-forced vibrations on post-tensioned concrete slabs by finite element techniques. In addition, the feasibility and accuracy of the finite element analysis is checked based on SCI and CSTR43 standard calculations. The influence of length, width, thickness and damping ratio on vibration comfort is investigated. The following conclusions are obtained from the two methods analysis:

- The three vibration evaluation parameters, natural frequency, peak acceleration and response factor, can simultaneously meet the mainstream international evaluation standards, and can be used as a comprehensive method for assessing the vibration serviceability of floor slabs;
- The error between the CSTR43/SCI standard calculations and the Strand7 results are within reasonable range, establishing the feasibility and accuracy of the Strand7 finite element analysis of vibration serviceability of post-tensioned concrete;
- Increasing span of a post-tensioned concrete slab will reduce the natural frequency. An increase in span reduces the peak acceleration and response factor within a certain range, but beyond that range leads to increased vibration. The span needs to be adjusted to meet the limits of the natural frequencies in the standard;
- Increasing the thickness of post-tensioned concrete slabs can improve the vibration service performance of the slab by increasing the fundamental frequency and reducing the peak acceleration and response factors;

## 7. Recommendations for Future Study

The following points need further improvement and research for vibration serviceability of post-tensioned concrete slabs:

- This paper analyses the vibration problems of simple support as slab-column connection methods, in the future the influence of different slab-column connection methods on vibration serviceability needs to be considered;
- This paper only uses Strand7 for the finite element analysis, which simplified some structural details due to technical limitations. Future modelling comparisons in other FEA software should be made to determine the appropriate FEA method;
- This paper only studies the vibration caused by a human pedestrian load at a specified point, the vibration caused by different movement frequencies and path choices of pedestrians needs further study;

- In this paper, the vibration source is only one-person pedestrian load, the problem of multiple people walking on a post-tensioned concrete slab needs to be further investigated;
- In this paper, only one subdivision size was considered in the finite element analysis. In future studies, multiple subdivision sizes are needed to investigate the effect of mesh sensitivity and to obtain more accurate results.
- Further experimental investigations are required to suggest a robust practical recommendations.

**Author Contributions:** Conceptualization, F.T., Z.H., Q.W. and W.M.; Software, F.T., Z.H., Q.W. and W.M.; Validation, F.T., Z.H. and W.M.; Formal analysis, Z.H., Q.W. and W.M.; Investigation, F.T., Z.H., Q.W. and W.M.; Resources, F.T., Z.H.; Data curation, W.M.; Writing—original draft, Z.H., Q.W. and W.M.; Writing—review & editing, F.T.; Supervision, F.T.; Project administration, F.T. All authors have read and agreed to the published version of the manuscript.

**Funding:** This research received no external funding.

**Institutional Review Board Statement:** Not applicable.

**Informed Consent Statement:** Not applicable.

**Data Availability Statement:** Not applicable.

**Conflicts of Interest:** The authors declare no conflict of interest.

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
