# Peer review of "Numerically Evaluation of Dynamic Behavior of Post-Tensioned Concrete Flat Slabs under Free Vibration"

_sustainability, doi:10.3390/su15010845_

Round 1

Reviewer 1 Report

This paper investigated the dynamic behaviour of post-tensioned concrete flat slabs with different geometries and damping ratios. Four groups of models with different lengths, widths, thicknesses and damping ratios designed according to the AS3600 standard. The vibration assessment parameters were  used as natural frequency, peak acceleration, and response factor. Both the SCI/CSTR43 standard theoretical calculations method and the Strand7 finite element analysis (FEA) method are used to determine the effect of different geometries and damping ratios on vibration.

The paper is interesting and it can be published after the following revisions.

1.       Different parts of introduction section must be combined, and the authors must delete the general information in these parts. Also, the authors must improve the literature review by discussion the recent works.

2.       In methodology, “400*400 m column”, is it true?

3.       The authors should present their mesh sensitivity results?

4.       The authors must justify their results not just saying what is the behavior of structure due to changing the parameters.

5.       Why the damping ration does not have effect on the frequencies? Is it true?

6.       The authors must verify their results with previous works by applying some simplifications, because their theoretical and FEM results show discrepancy in most of cases.

Author Response

Reviewer’s comments

Authors’ reply

 Different parts of introduction section must be combined, and the authors must delete the general information in these parts. Also, the authors must improve the literature review by discussion the recent works.

The format of the introduction has been revised, and some of the content has been removed and merged. Added some new literature to enhance the literature review.

   In methodology, “400*400 m column”, is it true?

Thank you for your comments. The column size used in the modelling and analytical calculations was 400*400mm. An error was made in collating the format and has been corrected in the article.

The authors should present their mesh sensitivity results?

The model is subdividing in each direction. More than 15 subdivisions were made in both directions of the slab. The area of each mesh is approximately 0.3 m2. The effect of mesh sensitivity on the results is not considered in this paper and will be presented and explained in 3.4. Limitation, 5. Discussion about the Results and 6. Recommendations for future study. Overall, we have had a computational limitation to do more mesh study investigations.

 The authors must justify their results not just saying what is the behaviour of structure due to changing the parameters.

Following the revision, in 5. Discussion about the Results, the experimental results obtained from the above study are analysed theoretically and compared with results from other literature. While explaining what is the behaviour of structure when changing the slab parameters, the reasons are explained and the results are justified based on the derivation of theoretical equations and comparison with other literature.

 Why the damping ration does not have effect on the frequencies? Is it true?

For natural frequencies, this is because the equation provided by CSTR43 (Page16 Equation 12) is not parameterised by the damping ratio, which is used in this paper to study the natural frequency. In the natural frequency solver of Strand7, the effect of damping ratio on the natural frequency is not considered. In the guidelines given by SCI and CSTR43, natural frequency as a physical property of the floor system is mainly influenced by material properties, shape, temperature, etc. The effect of damping ratio on natural frequency is also not considered. Therefore, for slab systems, higher damping ratio does not increase natural frequency.(Section 5.3 )

Smith, A.L., Hicks, S.J. and Devine, P.J., 2009. SCI P354: Design of floors for vibration: a new approach. The Steel Construction Institute, Ascot.

For forced vibrations frequencies, the increase in the damping ratio effectively reduces the peak acceleration and the corresponding factor due to the pedestrian load.

Therefore, in this paper, when damping ratio is changed, the natural frequency remains constant, but the peak acceleration and the response factor of the forced vibration are affected.

The authors must verify their results with previous works by applying some simplifications, because their theoretical and FEM results show discrepancy in most of cases.

In 5. Discussion about the Results, some comparisons with other results in the literatures are added, the differences in length and width from other experimental results are analysed and explained, presumably due to the small percentage of length and width changed, and the reasons for these differences are discussed.

Reviewer 2 Report

This paper aims to investigate the dynamic behavior of post-tensioned concrete flat slabs with different geometries and damping ratios. 

And four groups of models with different lengths, widths, thicknesses, and damping ratios according to the AS3600 standard were used to perform the numerical analysis and analyze the results.

However, the reviewer was very confused while reading the paper. This paper is not a dissertation. Every chapter should be written more concisely. In particular, chapters 1 and 2 describe the general content as too long.

The length of the slab, which is one of the parameters in this study, is 8 m, 10 m, and 12 m. By the way, are there any structures that use more than 10m of slab length?

Wouldn't it be more common to use girders if the slabs are going to be over 6m long? Is it really necessary to make slabs longer than 10m with post-tensioned concrete flat slabs?

The paper mentions:

"The error between the CSTR43/SCI standard calculations and the Strand7 results are within reasonable range, establishing the feasibility and accuracy of the Strand7 finite element analysis of vibration serviceability of post-tensioned concrete;"

It is considered that the method for deriving reasonable results is already presented in the standard. Is the method proposed in this study really necessary?

Author Response

Reviewer’s comments

Authors’ reply

Every chapter should be written more concisely. In particular, chapters 1 and 2 describe the general content as too long

The format of the introduction has been revised, and some of the content has been removed and merged. Added some new literature to enhance the literature review.

The length of the slab, which is one of the parameters in this study, is 8 m, 10 m, and 12 m. By the way, are there any structures that use more than 10m of slab length?

In some columns free layout, there is possibility to design span more than 14 m in the PT concrete slabs.

Specially, when it comes to designing show rooms, there is more interest in constructing 14-16 m spans.

Wouldn't it be more common to use girders if the slabs are going to be over 6m long? Is it really necessary to make slabs longer than 10m with post-tensioned concrete flat slabs?

This is a PT concrete slab layout. It is not a composite steel concrete beam floor systems that we can add more girders.

In the PT slabs, either you may change the geometry of the band beams or the concrete slabs.

The paper mentions:

"The error between the CSTR43/SCI standard calculations and the Strand7 results are within reasonable range, establishing the feasibility and accuracy of the Strand7 finite element analysis of vibration serviceability of post-tensioned concrete;"

It is considered that the method for deriving reasonable results is already presented in the standard. Is the method proposed in this study really necessary?

Thanks for your comments, the current method is a novel method to cover the main limitation of the current standard when it comes to designing non -standard spans. It provides a significant technique that practical engineers can follow it up when there is a critical limitations in the available code of practice.

Reviewer 3 Report

Dear Authors,

The main objective of the article is to investigate the effect of post-tensioned concrete flat slab geometry and damping ratios on vibration comfort using finite element analysis. Post-tensioned concrete flat slab models of different lengths, widths, thicknesses, and damping ratios are designed using the AS3600-2018 specification and analyzed by finite element modelling using Strand7. These were used to determine the influence of each parameter on the vibration serviceability by comparing the control variable method with the reference model. The vibration assessment parameters were used as natural frequency, peak acceleration, and response factor.

The main conclusions are that the Strand7 FEA method is highly accurate and feasible. The span in both directions has a large effect on the natural frequency and increasing both the slab thickness and the damping ratio are effective methods to improve the vibration serviceability. Based on the research in this paper, recommendations are provided for future vibration design of post-tensioned concrete slabs.

The article is well presented with an interesting contribution to the research area;
The abstract as well as the introduction are well presented. The objectives are well defined.
The figures and graphs are well presented;
The results were clearly presented and well interpreted by the authors;
The conclusions are well summarized and corroborate the presented objectives. Some recommendations for future study are also presented.
Overall, I understand that the article is well written. The experimental work will always be interesting for professionals in the field in the context presented in the article.

Author Response

Reviewer’s comments

Authors’ reply

The experimental work will always be interesting for professionals in the field in the context presented in the article.

I would like to deeply appreciate your professional comments. However, doing experimental investigations are required to have adequate fund as well as relevant spaces. They are significant limitations and we will do our best to consider it for the future studies.

Round 2

Reviewer 1 Report

The paper is revised according to my comment and can be accepted in the present form.

Author Response

Thanks for your kind consideration. 

Reviewer 2 Report

Thank you for responding to the review. Your manuscript is concise and easy to understand compared to previous manuscripts. However, reviewers are still concerned about the slab's length. This is a safety issue. This problem cannot be solved using numerical methods alone. It is believed that this problem needs to be continuously reviewed through many construction cases.

리뷰에 답변해 주셔서 감사합니다. 귀하의 원고는 이전 원고에 비해 간결하고 이해하기 쉽습니다. 그러나 리뷰어들은 여전히 ​​슬래브의 길이에 대해 우려하고 있습니다. 이것은 안전 문제입니다. 이 문제는 수치적 방법만으로는 풀 수 없습니다. 이 문제는 많은 시공 사례를 통해 지속적으로 문제점을 검토해야 할 것으로 사료된다.

Author Response

Reviewer’s comments

Authors’ reply

Thank you for responding to the review. Your manuscript is concise and easy to understand compared to previous manuscripts. However, reviewers are still concerned about the slab's length. This is a safety issue. This problem cannot be solved using numerical methods alone. It is believed that this problem needs to be continuously reviewed through many construction cases.

Once again, thanks so much for your technical comments. In the current simulation and technical assessment, the main objectives of the current study are to evaluate the dynamic behaviour of the post-Tension concrete slab under free vibration. Thus, this study can be classified as a part of the serviceability limit state. Therefore, it is not part of the ultimate limit state which is directly relevant to structural assessment of the failure and post failure of the PT slabs. We can conclude that focusing on technical designing of the PT slabs under ultimate loading which can be also concerned with checking safety requirement is out of scope of the current study. In my opinion, it needs to allocate separate research to address the ultimate limit state issue in designing PT slabs subjected to different type of loadings. Overall, this study was devoted on the serviceability limit state.    
